# Chemical profile and analysis of biosynthetic pathways and genes of volatile terpenes in *Pityopsis ruthii,* a rare and endangered flowering plant

Xinlu Chen[1], Marcin Nowicki[2]*, Phillip A. Wadl[3], Chi Zhang[1], Tobias G. Köllner[4], Miriam Payá-Milans[2¤], Matthew L. Huff[2], Margaret E. Staton[2], Feng Chen[1], Robert N. Trigiano[2]

**1** Department of Plant Sciences, University of Tennessee, Knoxville, TN, United States of America, **2** Department of Entomology and Plant Pathology, University of Tennessee, Knoxville, TN, United States of America, **3** United States Department of Agriculture, Agricultural Research Service, U. S. Vegetable Laboratory, Charleston, SC, United States of America, **4** Department of Biochemistry, Max Planck Institute for Chemical Ecology, Jena, Germany

¤ Current address: Computational Systems Medicine, Institute of Biomedicine of Seville (IBiS), Hospital Virgen del Rocío, Sevilla, Spain

* mnowicki@utk.edu

⊙ OPEN ACCESS

**Data Availability Statement:** All relevant data are within the paper and its Supporting Information files. The sequences of 5 functionally characterized

## Abstract

It is critical to gather biological information about rare and endangered plants to incorporate into conservation efforts. The secondary metabolism of *Pityopsis ruthii,* an endangered flowering plant that only occurs along limited sections of two rivers (Ocoee and Hiwassee) in Tennessee, USA was studied. Our long-term goal is to understand the mechanisms behind *P. ruthii*'s adaptation to restricted areas in Tennessee. Here, we profiled the secondary metabolites, specifically in flowers, with a focus on terpenes, aiming to uncover the genomic and molecular basis of terpene biosynthesis in *P. ruthii* flowers using transcriptomic and biochemical approaches. By comparative profiling of the nonpolar portion of metabolites from various tissues, *P. ruthii* flowers were rich in terpenes, which included 4 monoterpenes and 10 sesquiterpenes. These terpenes were emitted from flowers as volatiles with monoterpenes and sesquiterpenes accounting for almost 68% and 32% of total emission of terpenes, respectively. These findings suggested that floral terpenes play important roles for the biology and adaptation of *P. ruthii* to its limited range. To investigate the biosynthesis of floral terpenes, transcriptome data for flowers were produced and analyzed. Genes involved in the terpene biosynthetic pathway were identified and their relative expressions determined. Using this approach, 67 putative terpene synthase (*TPS*) contigs were detected. TPSs in general are critical for terpene biosynthesis. Seven full-length TPS genes encoding putative monoterpene and sesquiterpene synthases were cloned and functionally characterized. Three catalyzed the biosynthesis of sesquiterpenes and four catalyzed the biosynthesis of monoterpenes. In conclusion, *P. ruthii* plants employ multiple TPS genes for the biosynthesis of a mixture of floral monoterpenes and sesquiterpenes, which probably play roles in chemical defense and attracting insect pollinators alike.

terpene synthases reported in this paper have been deposited in the GenBank (https://www.ncbi.nlm. nih.gov/genbank/) database (accession numbers ON166544 - ON166550). The 3 transcriptome archives are available under NCBI PRJNA778727 (https://www.ncbi.nlm.nih.gov/bioproject/?term= PRJNA778727).

**Funding:** This work was supported by the U.S. Department of Agriculture (USDA/MOA number 58-6404-1-637) awarded to RNT. Mention of trade names or commercial products in this article is solely for the purpose of providing specific information and does not imply recommendation by the U.S. Department of Agriculture or the authors. USDA is an equal opportunity provider and employer. The publication costs are covered by the University of Tennessee Open Publishing Fund, awarded to RNT. The funding agencies played no role in the study design, data collection and analysis, decision to publish, or preparation of the manuscript.

**Competing interests:** The authors have declared that no competing interests exist.

## Introduction

Conserving rare and endangered plant species is a daunting task. The passage of the Endangered Species Act in 1973 in the USA provided the legal mandate for collecting and analyzing biological information for such plants [1]. This study focused on *Pityopsis ruthii* Small, a narrowly distributed endangered plant that occurs only along small sections of the Hiwassee (~ 4 km) and Ocoee (~ 2.5 km) Rivers in Polk County, Tennessee, USA. This small, fall-flowering, herbaceous species belongs to the Asteraceae and typically grows up to 30 cm in height within cracks of massive phyllite boulders situated between the river channel and adjacent forested slopes. *P. ruthii* is a small, rounded perennial with erect green to brown-colored stems that are covered with silvery hairs and yellow daisy-like flowers [2].

Although *P. ruthii* can persist in shaded habitats, flowering, seed set, and establishment of plants is the most successful in open areas that receive full sun for a substantial portion of the day [3, 4]. White [4] examined the effects of light intensity, drought, rock surface, soil, leaf, and air temperatures on the growth of *P. ruthii* and of competing species and demonstrated the ability of *P. ruthii* to survive the extremes of the harsh environment. *P. ruthii* is tolerant of prolonged drought [3] and of inundating high flow events (A. J. Dattilo, *unpublished*), but the plant is a very poor competitor against other species when grown in areas of deeper soils that build-up on the boulder complex or outside of the exposed boulder complexes [5, 6]. Regardless, the species has a very narrow range in a habitat that is subjected to extreme conditions. Although *P. ruthii* is well adapted to the harsh environment, the range and combination of possible adaptations that permit it to survive the intermittent, but often prolonged, drought and submergence are not known. Although extreme drought and flood events are increasing globally, certain ecological niches have historically undergone regular, radical shifts, and a few plant species, such as *P. ruthii*, have adapted to survive such events.

*P. ruthii* is self-incompatible and produces scant amounts of pollen per flower. It is dependent on insects for cross-pollination. Honeybees, bumblebees, and -to a lesser extent- native bees are the primary pollinators for the species [3, 7]. For successful cross-pollination, these bees must find small populations in nature, and therefore, a chemical attractant may provide signals for insects [8]. Considering the peculiar ecological niche and the biology of *P. ruthii*, it is important to understand the mechanisms that make it a successful plant in this environment.

Many such mechanisms contribute to the adaptation of plants to unique niches. One of the possible mechanisms is the production of a myriad of secondary metabolites, which are characterized by their vast chemical diversity. Terpenes are the largest family of secondary metabolites produced by plants and can be further categorized as monoterpenes (C10), sesquiterpenes (C15), and diterpenes (C20) [9]. The biological functions of terpenes include tolerance of abiotic stresses such as flooding and drought, defense against insects and microbial pathogens, habitat acclimation, and attraction of beneficial organisms such as pollinators [8–14]. Mechanistically, terpenes can regulate the fluidity of cell membranes, maintain membrane integrity, and stabilize photosynthetic machinery under stress conditions [12].

Our long-term general goal was to understand the mechanisms underlying the adaptation of *P. ruthii* to very limited areas in Tennessee. In this study, we focused on the analyses of secondary metabolites from various tissues, particularly flowers. With terpenes identified as the main constituents of secondary nonpolar metabolites from flowers of *P. ruthii*, it was also our objective to use transcriptomic and biochemical approaches to understand the genomic and molecular basis of terpene biosynthesis in *P. ruthii* flowers.

## Materials and methods

### Plant material

Stem cuttings of *P. ruthii* plants collected from populations growing along the Hiwassee and Ocoee Rivers, Tennessee, were rooted and grown in pots under natural light [15]. Plants were collected under Tennessee Valley Authority Permit # TE117405-2 and U.S. Fish and Wildlife Service Permit # TE134817-1. Roots, stems, undamaged leaves, and fully open flowers were harvested in September 2015 from three-year-old plants growing in Pro-Mix BX (Premier Tech Horticulture, Quakertown, PA, USA) maintained outdoors at the University of Tennessee. Leaves were cut longitudinally into five sections using a sterile scalpel for physical wounding. Fully open flowers were used to collect volatile terpenes.

### Chemical profiling

Leaves, stems, roots, and rhizomes were collected from plants and ground in liquid nitrogen to a fine powder with mortar and pestle. Methylene chloride containing 0.003% (v/v) of 1-octanol as an internal standard was added to 100 mg of each tissue type at a ratio of 10:1 (v/w). Materials were extracted for 4 h with continuous shaking at 100 rpm at room temperature. After centrifugation at $13000 \times g$ for 30 min, 5 μL of extract from each preparation was analyzed using Shimadzu QP5050A Gas chromatography-Mass spectrometry (GCMS). Organic extraction of each tissue type was analyzed in triplicate with each replicate combining plant materials collected from three individual plants. For GCMS analyses, a splitless injection port with injection temperature 250°C was set and the column temperature was set at 60°C with 6 min holding, then increased to 280°C at a rate of 5°C per min. Separation was carried out on a Rxi®-5Sil MS column (30 m × 0.25 mm i.d. × 0.25 μm thickness, Restek, Bellefonte, PA, USA). Helium as carrier gas was set at flow rate of 1 mL min$^{-1}$. Terpenes were identified using various available MS libraries (NIST, WILEY, and Adams) and by comparison to the mass spectra of authentic standard compounds (https://www.sigmaaldrich.com). Quantification was performed by peak area comparison to that of the internal standard.

Volatiles from *P. ruthii* were collected using an open headspace sampling system (Analytical Research System, Gainesville, FL) as previously described [16]. Respective plant materials (about 1 to 2 grams per sample) were placed in a 40-mL glass beaker filled with 10 mL of sterile distilled water. The glass beaker with plant samples was placed in a glass chamber with an air inlet and outlet. With the flow of air, volatiles emitted from plant samples were trapped on the volatile collection trap (VCT) containing 25 mg Porapak-Q$^{TM}$ (http://www.volatilecollectiontrap.com/) placed at the air outlet. After 4 h of collection at room temperature, volatiles were eluted using 100 μL of methylene chloride containing 1-octanol (0.003% v/v) as internal standard and then directly injected into GC-MS for analysis, as described above. Each headspace collection was replicated three times with each replicate combining appropriate plant materials collected from three individual plants. Terpenes were identified and quantified as described.

### Transcriptome sequencing and analysis

To gain access to the *P. ruthii* transcriptome, total RNA was isolated from freshly collected flowers that were immediately flash frozen in liquid nitrogen. Three independent accessions maintained at the University of Tennessee were sampled, and the florets removed, so that only involucre tissues were used. About 100 mg of tissue per accession were subjected to RNA isolation using the Ribospin II kit (GeneAll Biotechnology, Seoul, Korea) following the manufacturer's protocol. Total RNA quality was assessed using BioAnalyzer RNA chip (Agilent, Santa

Clara, CA, USA), after which the RNA was sequenced using HiSeq Illumina (150 bp PE; Gene-Wiz Inc., South Plainfield, NJ, USA).

Rcorrector [17] was applied to raw reads with default parameters. Rcorrector is a kmer-based error correction method that uses a De Bruijn graph to represent trusted k-mers. This method was similar to that used on *de novo* assembly and helped improve the quality of assemblies. Corrected reads were trimmed to remove the Illumina adapter sequences using Skewer v0.2.2 [18], using a minimum read length cutoff of 30 bp. FastQC v0.11.4 (http://www.bioinformatics.babraham.ac.uk/projects/fastqc/) was used for quality control of reads. Reads matching ribosomal, plastid, and mitochondrial DNA were removed from the analysis using Bowtie2 [19].

Cleaned reads were used to assemble transcripts with Trans-ABySS [20] on a multi-kmer assembly, and used kmers of 25, 45, and 65 for individual assemblies with minimum contig length of 200 bp, and transabyss-merge to combine them. Substantial removal of assumed isoforms was carried-out afterwards with RapClust [21], which groups transcripts using information from multi-mapper paired-ended reads. Read mapping was performed with Salmon v0.8.2 [22], a fast quasi-mapping tool. The clustering information yielded by RapClust was used to obtain a reduced transcriptome after the selection of the longest transcript per cluster.

Basic assembly metrics were obtained with in-house script. Completeness of the *de novo* transcriptome assembly was assessed with BUSCO v2.0 [23]. A custom protein database of 54 enzymes involved in terpene biosynthesis was obtained from their EC codes with KEGGREST [24]. A custom database for the *ACT*, *EF1-A*, *GAPDH*, and *COX1* housekeeping genes from UniProtKB in eudicots was used to find putative sequences in *P. ruthii*. Read counts per contig for the extended *P. ruthii* TPS candidate list were normalized taking into account the respective contig lengths (RPKMi [25]).

### Identification of terpene synthase genes in *Pityopsis ruthii*

Protein coding regions of the flower transcriptome were predicted using Transdecoder 5.5.0 [26]. HMM files with N-domain (PF01397) and C-domain (PF03936) downloaded from (http://pfam.xfam.org/) were used as a query to identify TPSs using HMMER v3.3.1 [27] with an E-value of $1e^{-5}$. Multiple sequence alignment of putative *PrTPS* genes and TPS family of *Arabidopsis thaliana* [28] was performed using MAFFT v7.480 [29] with 1,000 iterate improvement. Maximum likelihood tree was built using MEGA v11.0 [30] with the JTT model and 1,000 bootstraps. Only bootstrap support values higher than 50% were shown in the tree.

### *TPS* genes cloning and terpene synthase activity assays

Total RNA was isolated from fully open flowers using the RNeasy Plant Mini Kit (https://www.qiagen.com/) according to the manufacturer's protocol. cDNA was synthesized using a First Strand cDNA Synthesis Kit (https://www.cytivalifesciences.com/) according to the manufacturer's protocol. Primer Notl-(dt)18 (5′ -AACTGGAAGAATTCGCGGCCGCAGGAA TTTTTTTTTTTTTTTTT-3′) was used for cDNA synthesis. Signal peptides were predicted using TargetP [31], and were cleaved before the conserved RRWx8W motif to create truncated recombinants, which typically improves activity of the soluble enzymes [32]. Full length cDNAs of PrTPS1, PrTPS2, PrTPS3, PrTPS4, PrTPS5, PrTPS6, and PrTPS7 obtained from the transcriptome analyses were amplified using gene specific primers extended with restriction enzyme sites (Table 1) and cloned into pGEM®-T Easy Vector (www.promega.com). After sequencing confirmation, they were cut with restriction enzymes and cloned into pET32a. The plasmids containing full length cDNAs were transformed into *E. coli*

**Table 1. Primers used for cloning cDNA in this study.**

| Gene name | Primers | Sequences[a] |
|---|---|---|
| PrTPS1 | Forward | 5'-ggatccATGAGACGATCGGCAAACTATGAC-3' |
| | Reverse | 5'-gaattcTACTTTACTCCATGGATTGGATTG-3' |
| PrTPS2 | Forward | 5'-gagctcATGAGACGATCAGCAAACCATC-3' |
| | Reverse | 5'-gcggccgcTTAGTGATCCCCTAATTTAGGTTTAG-3' |
| PrTPS3 | Forward | 5'-ggatccATGAGAAGATCAGCAAATTGGCAAC-3' |
| | Reverse | 5'-aagcttAGATTGGATTAACAAACAATGATAATG-3' |
| PrTPS4 | Forward | 5'-ggatccATGTACACAAACATCGATCAATG-3' |
| | Reverse | 5'-gagctcCTACACTTCAATCCTATAGGAAATG-3' |
| PrTPS5 | Forward | 5'-gagctcATGTCAACTTTGCCAGTTTCTATTG-3' |
| | Reverse | 5'-ctcgagTTAAATAACCATAGGGCGAACA-3' |
| PrTPS6 | Forward | 5'-gagctcATGGCTGCTAAACAAGGAGATCTTATTC-3' |
| | Reverse | 5'-ctcgagTCAAACACTCATAACATTAACGAAAC-3' |
| PrTPS7 | Forward | 5'-gagctcATGGCTGCTGCTCAACATGGAG-3' |
| | Reverse | 5'-ctcgagTCAAACACTCATAGCATTAACG-3' |

[a]The sequences for restriction sites were added to 5' sites: ggatcc—*Bam*HI, gaattc—*Eco*RI, gagctc—*Sac*I, gcggccgc—*Not*I, aagctt—*Hind*III, ctcgag–*Xho*I.

BL21 codon plus (DE3) for heterologous expression. *E. coli* harboring the expression plasmids were cultured at 37°C to an $OD_{600}$ of 0.6, then expression was induced through addition of 1 M isopropyl-1-thio-D-galactopyranoside (IPTG) to liquid LB cultures until final concentration of 1 mM. The cells were harvested by centrifugation at $4000 \times g$ after 16 h of culture at 22°C. Cell pellets were resuspended in chilled extraction buffer (50 mM MOPSO, pH 7.0, 5 mM $MgCl_2$, 5 mM DTT, 5 mM Na-Ascorbate, 0.5 mM PMSF, 10% (v/v) glycerol) and disrupted by ultrasonication $6 \times 30$ s using cell disruptor (Misonix, Framingdale, NY). Supernatant obtained after centrifugation at $13000 \times g$ to remove cell debris was desalted into assay buffer (10 mM MOPSO, pH 7.0, 1 mM dithiothreitol, 10% (v/v) glycerol) by passage through a PD-10 desalting column (http://www.cytivalifesciences.com/). Enzyme assays were performed in Teflon-sealed screw capped 2 mL glass vials containing 50 μL assay buffer (20 mM MOPSO, pH 7.0, 10 mM $MgCl_2$, 1 mM DTT, 0.2 mM $NaWO_4$, 0.1 mM NaF, 0.05 mM $MnCl_2$), 50 μL desalted crude protein extract, and 4 μM FPP or GPP, respectively. Volatiles produced by each respective reaction were collected by solid-phase microextraction (SPME, https://www.sigmaaldrich.com/) for 30 min at 22°C and analyzed using GC-MS. Terpenes were identified as described previously [33]. As a negative control, the empty vector pET32a was introduced into *E. coli* Bl21 and the crude protein extracts were isolated and expressed with the substrates GPP, FPP, respectively, and analyzed as described above.

## 3-D modelling

The predicted translated amino acid (AA) sequences of PrTPS6 and PrTPS7 were analyzed in 3-dimensional models, owing to their very close similarity (Fig 3; S1 Fig). The sequences were aligned and projected using the Swiss Model server and their internal implemented software [34]. The display of 3-D structure of both coding sequences simultaneously was arranged to highlight the differences (red hue) vs. the AA identity (green hue) in the Swiss Model server over a web browser. Highlights of the identified conserved motifs were marked in MS PowerPoint.

## Statistical analyses

All experiments were performed in at least three technical replicates, unless stated otherwise. Significance was calculated in one- or two-way ANOVA, respectively, including the analyses of factorial interaction. Post-hoc test of Fisher's Honestly Significant Difference was carried-out at α = 0.05. All these were computed using R v4.2.2 and the package *agricolae* v1.3–5 [35].

# Results

## Terpene profiles in leaves, stems, flowers, and roots of *Pityopsis ruthii* plants

The volatile terpene chemistry of roots and rhizomes was assessed using organic extraction and the terpenoid compounds were identified using GC-MS. Two sesquiterpenes, β-elemene and allo-aromadendrene were detected from root tissues with concentrations of 14.14 μg × g $FW^{-1}$ and 0.28 μg × g $FW^{-1}$, respectively (Table 2). For the above-ground parts, stems of *P. ruthii* produced allo-aromadendrene at relatively low concentration of 0.80 μg × g $FW^{-1}$ (Table 2). Six terpenes were detected in leaves and included two monoterpenes and four sesquiterpenes. Despite having a higher number of individual terpenes, the observed sum concentration of all terpenes in leaves was higher than that in stems (Table 2). Twelve terpenes including 4 monoterpenes and 8 sesquiterpenes were identified in the flower extracts (Table 2). Among them, the most abundant terpenes were two sesquiterpenes β-elemene (14.05 μg × g $FW^{-1}$) and allo-aromadendrene (16.42 μg × g $FW^{-1}$) that together accounted for 54.20% of total terpenes detected in the *P. ruthii* flowers extracts. Among four monoterpenes, the concentrations of myrcene (5.13 μg × g $FW^{-1}$) and limonene (7.09 μg × g $FW^{-1}$) were higher than those of α-pinene and β-pinene, accounting for 9.12% and 12.61% of total terpenes detected in the *P. ruthii* flower extracts, respectively. Except for allo-aromandendrene (16.42 μg × g $FW^{-1}$) and β -elemene (14.05 μg × g $FW^{-1}$), all other remaining sesquiterpenes

**Table 2. Concentrations (μg × g $FW^{-1}$) of extracted terpenes from roots, stems, and leaves of *Pityopsis ruthii*.**

| Compound[1] | Flowers[24A] | Leaves[B] | Stems[B] | Roots[B] | Rhizomes[B] |
|---|---|---|---|---|---|
| α-pinene | 2.1±0.020[FGH] | -[3] | - | - | - |
| β-pinene | 1.03±0.10[IJK] | - | - | - | - |
| myrcene | 5.13±0.15[D] | 1.37±0.18[HIJ] | - | - | - |
| limonene | 7.09±0.60[C] | 1.21±0.13[HIJK] | - | - | - |
| β-elemene | 14.05±0.55[B] | 3.25±0.10[E] | - | 14.14±1.37[B] | - |
| (*E*)-α-bergamotene | 1.98±0.11[GHI] | 2.49±0.56[EFG] | - | - | - |
| unknown sesquiterpene 1 | 1±0.09[IJK] | - | - | - | - |
| guaia-1(10),11-diene | 0.96±0.07[IJK] | - | - | - | - |
| unknown sesquiterpene 2 | 1.01±0.07[IJK] | - | - | - | - |
| aromadendrene | 0.76±0.07[JK] | - | - | - | - |
| allo-aromadendrene | 16.42±0.82[A] | 3.05±0.28[EF] | 0.18±0.04[K] | 0.28±0.10[K] | 0.21±0.03[K] |
| α-selinene | 4.68±0.31[D] | 0.66±0.10[JK] | - | - | - |

[1]: compound identified based on commercial database and the authentic compound standards

[2]: μg × g $FW^{-1}$ represents μg per gram fresh weight, values represent mean of three replicates ± standard deviation

[3]: not detected

[4]: For organs–single factorial ANOVA; same letters signify no statistical differences at α = 0.05 using post-ANOVA Fisher's Honestly Significant Difference. Critical Value of Studentized Range: 4.69. Minimum Significant Difference: 3.62. For organs and compounds: Two-way factorial ANOVA; same letters signify no statistical differences at α = 0.05 using post-ANOVA Fisher's Honestly Significant Difference. Critical Value of Studentized Range: 5.93. Minimum Significant Difference: 1.06.

were detected in low concentrations ($< 5\ \mu g \times g\ FW^{-1}$). Flowers of *P. ruthii* were the primary organs that produced and stored volatile terpenes.

## Flowers of *Pityopsis ruthii* emit a bouquet of volatile terpenes

Fully open flowers of *P. ruthii* are moderately fragrant to the human nose. To determine the chemical composition of their fragrance, volatiles emitted from the open flowers were collected using headspace collection and analyzed using GC-MS. Four monoterpenes, ten sesquiterpenes, and two diterpenes were detected from the flowers (Fig 1). The collective emission rate of volatile terpenes was $1632.74\ ng \times h^{-1} \times g\ FW^{-1}$. The four major monoterpenes accounted for 68% (w/w) of all terpenes emitted; comparatively, the sesquiterpenes were emitted in lower levels of less than $50.00\ ng \times h^{-1} \times g\ FW^{-1}$ (Table 3). α-Pinene was the most abundant volatile terpene emitted by flowers and accounted for almost one-half of the monoterpenes emitted but was undetected in leaves, stems, and roots. The second most

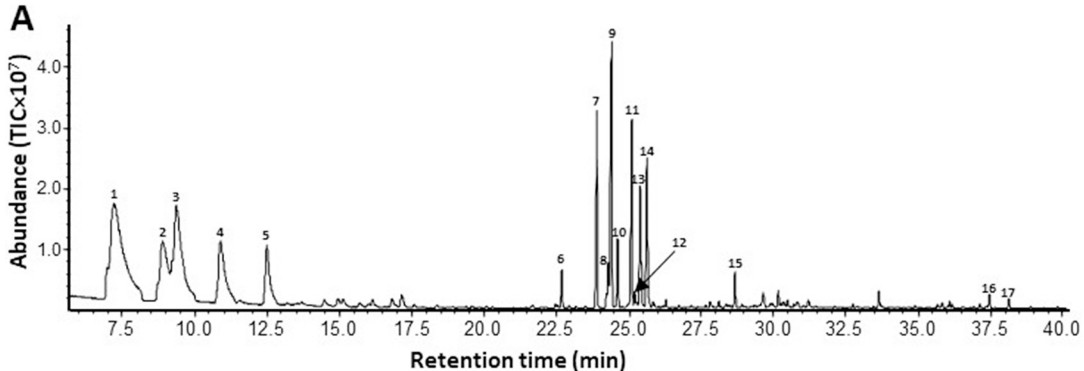

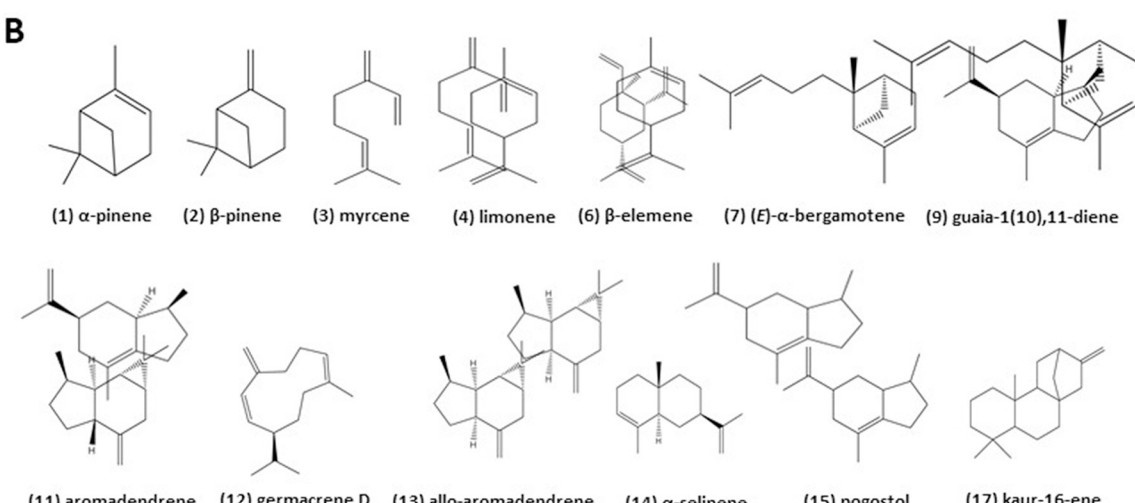

**Fig 1. Gas chromatograms of terpene volatiles emitted from open flowers and leaves of *Pityopsis ruthii*.** A. Compounds collected from open flowers using headspace. 1. α-pinene; 2. β-pinene; 3. myrcene; 4. limonene; 5 internal standard (1-octanol); 6. β-elemene; 7. (*E*)-α-bergamotene; 8. unidentified sesquiterpene 1; 9. guaia-1(10),11-diene; 10. unidentified sesquiterpene 2; 11. aromadendrene; 12. germacrene D; 13. allo-aromadendrene; 14. α-selinene; 15. pogostol; 16. unidentified diterpene 3; 17. kaur-16-ene. B. chemical structures of known compounds identified from open flowers in *Pityopsis ruthii*.

**Table 3. Terpene emission from *Pityopsis ruthii* flowers and leaves using headspace sampling (ng × h$^{-1}$ × g FW$^{-1}$).**

| Compound[1] | Emission rate | |
|---|---|---|
| | Flowers[24A] | Leaves[B] |
| α-pinene | 232.28±21.12[A] | –[3] |
| β-pinene | 90.27±2.92[B] | - |
| myrcene | 21.97±1.88[DEF] | - |
| limonene | 25.78±1.00[DE] | 0.81±0.21[G] |
| β-elemene | 10.29±2.80[EFG] | - |
| (*E*)-α-bergamotene | 23.41±0.97[DE] | - |
| unknown sesquiterpene 1 | 4.04±2.02[G] | - |
| guaia-1(10),11-diene | 41.92±3.14[C] | - |
| unknown sesquiterpene 2 | 7.34±0.31[FG] | - |
| aromadendrene | 24.87±1.87[DE] | - |
| germacrene D | 7.15±0.70[FG] | - |
| allo-aromadendrene | 22.51±2.52[DEF] | - |
| α-selinene | 26.17±1.67[D] | - |
| pogostol | 3.99±0.14[G] | - |
| unknown diterpene 1 | 0.96±0.12[G] | - |
| kaur-16-ene | 1.3±0.03[G] | - |

[1]: compound identified based on commercial database and authentic compounds

[2]: ng × h$^{-1}$ × g FW$^{-1}$ represents ng per hour per gram fresh weight; values represent the mean of three replicates ± standard deviation

[3]: not detected.

[4]: For organs–single factorial ANOVA; same letters signify no statistical differences at α = 0.05 using post-ANOVA Fisher's Honestly Significant Difference. Critical Value of Studentized Range: 4.69. Critical Value of Studentized Range: 2.81. Minimum Significant Difference: 16.12. For organs and compounds:: Two-way factorial ANOVA;; same letters signify no statistical differences at α = 0.05 using post-ANOVA Fisher's Honestly Significant Difference. Critical Value of Studentized Range: 5.56. Minimum Significant Difference: 15.12.

abundant terpene was β-pinene. Similar to the extracts from leaves, sesquiterpenes were common in the flower volatiles. The most abundant sesquiterpenes were (*E*)-α-bergamotene, guaia-1(10), 11-diene, α-selinene, aromadendrene, allo- aromadendrene, and (*E*)-α-bergamotene. Emission rates of the remaining sesquiterpenes were ≤ 10.3 ng × h$^{-1}$ × g FW$^{-1-1}$ and varied from about 0.96 to 10.29 ng × h$^{-1}$ × g FW$^{-1}$.

## Transcriptomic analyses of *Pityopsis ruthii* flowers

Sequencing of *P. ruthii* flower samples from three accessions yielded 191 million paired-end 150-bp Illumina reads (NCBI PRJNA778727). Trimming retained 99.8% of total reads with an average length of 145.6 bp. After removal of ribosomal reads, 98.8% of total reads were used for *de novo* assembly. Initial assembly from Trans-ABySS yielded a transcriptome consisting of 702,209 contigs of average length 528 bp and N50 of 630. Considering the putative introduced redundancy incorporated from merging the results of individual assembly runs, and to greatly reduce the number of similar sequences, RapClust was used. The reduced assembly consisted of 129,317 contigs of average length of 934 bp and N50 of 1,151. Despite the reduced transcriptome representing 18.4% of the initial one, quality expressed as BUSCO completeness was only slightly affected; from 1,440 BUSCOs searched, complete BUSCOs were reduced from 1,243 to 1,215 after clustering, and fragmented BUSCOs increased from 79 to 85. These results underscore the high representation of conserved orthologs sequences in the *P. ruthii*

transcriptome, suggesting its high quality. In our reduced transcriptome, 300 sequences displayed homology with any of the 54 enzymes searched that are involved in terpene biosynthesis (S1 Table). The respective normalized read counts for those putative TPSs varied in the transcriptome (S2 Table).

Similar to other plants, two pathways are naturally involved in terpene synthesis in *P.ruthii*: the mevalonate (MVA) pathway located in the cytosol and methyl-erythritol phosphate (MEP/DOXP) pathway located in the plastids; both pathways produce two C5 isoprene precursors isopentenyl pyrophosphate (IPP) and its isomer dimethylallyl pyrophosphate (DMAPP). Geranyl pyrophosphate (GPP), farnesyl pyrophosphate (FPP) and geranylgeranyl diphosphate (GGPP), substrates for monoterpenes, sesquiterpenes and diterpenes, respectively, are synthesized by the condensation of IPP and DMAPP by Isoprenyl diphosphate synthases (IDS) (Jia and Chen, 2016) [36]. We investigated transcriptome-covered terpene pathways of *P. ruthii* (Fig 2) and assessed the relative expression of genes in both the MVA and the MEP/DOXP pathways from *P. ruthii* during flowering using the transcriptome analyses. Copy numbers (assumed isoforms) of AACT, HNGS, HMGR, MK, PMK, and PMD in the MVA pathway were 8, 6, 5, 2, 2, and 3, respectively. Whereas, copy numbers of DXS, DXR, CMS, CMK, MDS, HDS, and HDR in the MEP/DOXP pathway were 6, 2, 1, 2, 2, 4, and 3, respectively. The transcriptional levels of the initial steps in terpene backbone biosynthesis were relatively the highest: AACT, HMGS, and HMGR in the MVA pathway, and DXS and DXR in the MEP/DOXP pathway (Fig 2).

## Identifying putative terpene synthase genes in *Pityopsis ruthii*

The functional analyses of genes associated with terpene metabolism of *P. ruthii* flowers also leveraged the generated transcriptome data. The HMM files of Terpene_synth_N (PF01397) and Terpene_synth_C (PF03936) were used to search for the putative terpene synthases (TPSs) in the transcriptomes. In total, 130 unigenes were identified as putative TPS. After removal of the repeat sequences, 67 genes were identified as unique *TPSs* (S1 Table). Lengths of the translated proteins for these genes varied from 101 to 793 AA, and most of them were deemed partial open reading frames (ORFs) by comparison with confirmed *TPS* proteins from other plants that showed sequences longer than 540 AA. Other features supporting the claim of their full lengths were the presence of start and stop codons, and detection of α and β domains in TPSs-a, -b, and -g, or α, β, and γ domains in TPSs-c and TPSs-e/f. Whereas, partial ORFs contained only one of these domains. There were seven *TPS* genes that appeared to be full length and were designated *PrTPS1* to *PrTPS7* (GenBank accession numbers: ON166544 to ON166550). *PrTPS1*, *PrtPS2*, *PrTPS3*, *PrTPS4*, *PrTPS5*, *PrTPS6*, and *PrTPS7* encode proteins of 592, 593, 597, 587, 569, 550, and 548 AA in length, respectively. These seven *PrTPS* genes were analyzed phylogenetically with the *TPS* gene family of *Arabidopsis*. Based on this analysis, *PrTPS1* to *-3* were classified into the TPS-b subfamily, *PrTPS5* to *-7* into the TPS-a subfamily, and *PrTPS4* into the TPS-g subfamily, respectively (Fig 3). At the N-terminal position of *PrTPS1* to *-3*, a conserved motif $RRX_8W$ was identified, typical for the TPS-b subfamily, whereas no such motif was detected in *PrTPS4* from the TPS-g subfamily. At the C-terminal position of *PrTPS1* to *-7*, a conserved motif RDR was present, except *PrTPS4* displaying a variant RDQ. Also, the Asp-rich DDxxD motif was detected in *PrTPS1* to *-7*. The NSE/DTE motif was detected in *PrTPS1* to *-7* with a variant DDxxGxxxE in *PrTPS6* and *-7*, and DDxxSxxxE in *PrTPS4* (S1 Fig).

## Biochemical characterization of TPSs from *Pityopsis ruthii*

Based on the phylogenetic analysis, *PrTPS1*, *PtTPS2*, and *PtTPS3* of the TPS-b subfamily and *PrTPS4* of the TPS-g subfamily were predicted to encode monoterpene synthases based on

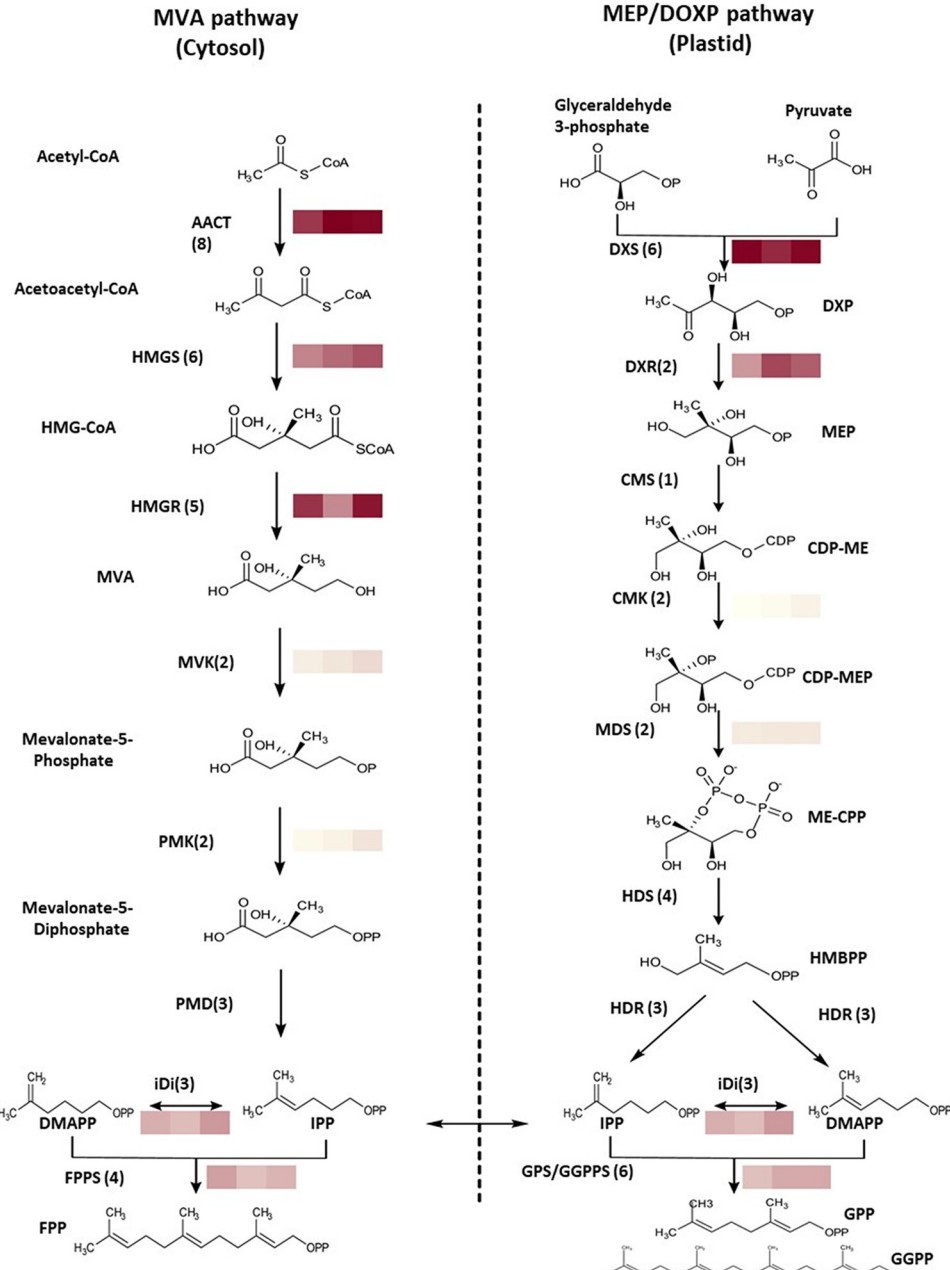

**Fig 2. Terpene pathways involved in terpene biosynthesis in *Pityopsis ruthii*.** Heatmap of the data from flower transcriptome analyses performed in triplicate along with representative gene indicated expression level, which was calculated with fragments per kilobase of transcripts per million mapped fragments (FPKM). Abbreviations of genes: *AACT*, acetoacetyl-CoA thiolase; *HMGS*, hydroxylmethylglutaryl-CoA synthase; *HMGR*, hydroxymethylglutaryl-CoA reductase; *MVK*, mevalonate kinase; *PMK*, 5-phospho-mevalonate kinase, *PMD*, mevalonate diphosphate decarboxylase; *FPPS*, farnesyl pyrophosphate synthase; *DXS*, 1-deoxy-ᴅ-xylulose-5-phosphate synthase; *DXR*, 1-deoxy-ᴅ-xylulose-5-phosphate reductoisomerase; *CMS*, 2-C-methyl-ᴅ-erythritol 4-phosphate cytidylyltransferase; *CMK*, 4-diphosphocytidyl-2-C-methyl-d-erythritol kinase; *MDS*, 2-C-methyl-D-erythritol 2,4-cyclodiphosphate synthase; *HDS*, (*E*)-4-hydroxy-3-methylbut-2- enyl diphosphate synthase; *HDR*, 4-hydroxy-3-methylbut-2-enyl diphosphate reductase; *IDI*, isopentenyl-diphosphate delta-isomerase; *GPPS*, geranyl diphosphate synthase. Compound abbreviations: HMG-CoA, 3-hydroxy-3-methylglutaryl-CoA; MVA, mevalonate; DXP, 1-Deoxy-D-xylulose 5-phosphate; MEP, 2-C-Methyl-D-erythritol 4-phosphate; CDP-ME, 2-C-Methyl-ᴅ-erythritol-2,4-cyclodiphosphate; CDP-MEP, 2-Phospho-4-(cytidine 5'-diphospho)-2-C-methyl-D-erythritol; ME-CPP, 2-C-methyl-ᴅ-erythritol-2,4-cyclodiphosphate; HMBPP, 1-hydroxy-2-methyl-2-(*E*)-butenyl-4-diphosphate; FPP, (*E,E*)-farnesyl pyrophosphate; IPP, isopentenyl pyrophosphate; DMAPP, dimethyallyl pyrophosphate; GPP, geranyl pyrophosphate; GGPP, geranylgeranyl pyrophosphate.

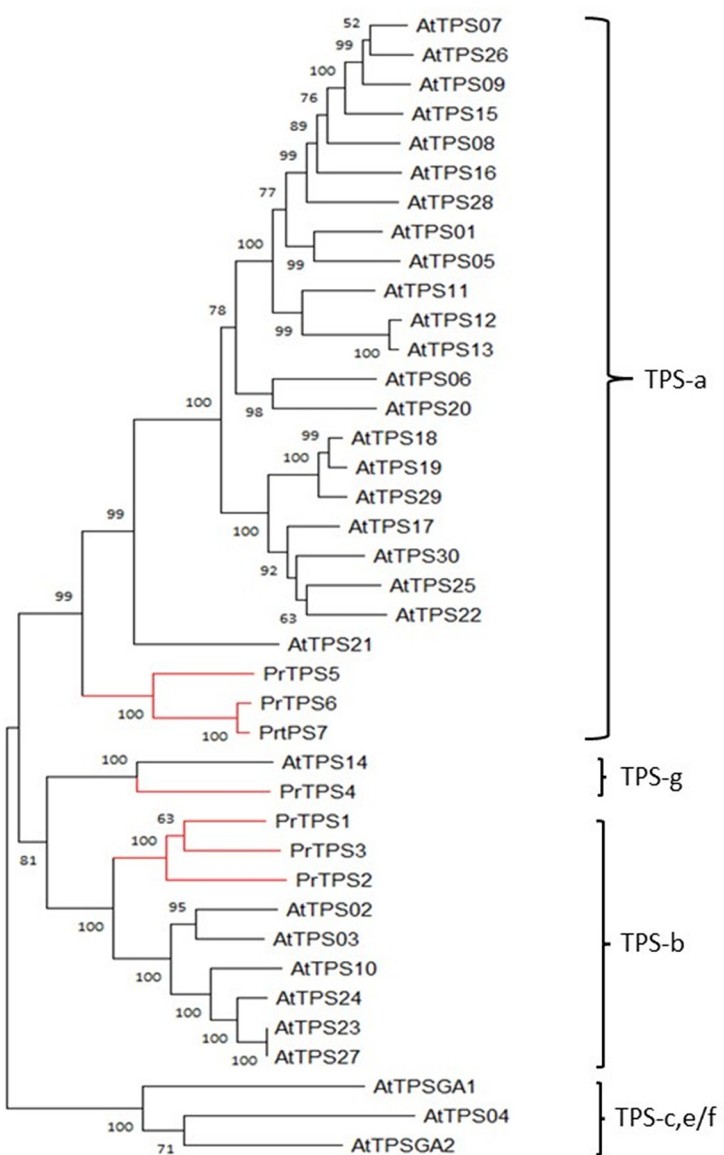

**Fig 3. Phylogenetic tree of full-length TPSs from *Pityopsis ruthii* (*PrTPS*s) and TPSs from *Arabidopsis thaliana* (*AtTPS*s).** Phylogenetic tree was reconstructed by maximum likelihood method based on JTT model. The classification of TPS subfamilies was determined as previously reported [9].

previous analysis of the *TPS* gene family [9] Recombinant proteins expressed in *E. coli* for each of these four *PrTPS* genes were tested for monoterpene synthase activities using GPP as substrate. Both *PrTPS1* (Fig 4A) and *PrTPS3* (Fig 4B) produced α-pinene as a major product and β-pinene as a minor one using GPP as substrate. Neither *PrTPS2* nor *PrTPS4* showed any activity under the conditions tested.

*PrTPS5*, *PrTPS6*, and *PrTPS7* of the TPS-a subfamily (Fig 3) were predicted to encode sesquiterpene synthases. These three genes were also expressed in *E. coli* to produce recombinant proteins, which were subject to sesquiterpene synthase activity assays. Recombinant PrTPS5 catalyzed the conversion of FPP into a single terpene, (*E*)-α-bergamotene (Fig 5A), which was one of the predominant sesquiterpenes identified in flowers (Fig 1A). PrTPS6 (Fig 5B) converted (*E*,*E*)-FPP to multiple sesquiterpenes with β-ylangene, γ-elemene, and germacrene D as

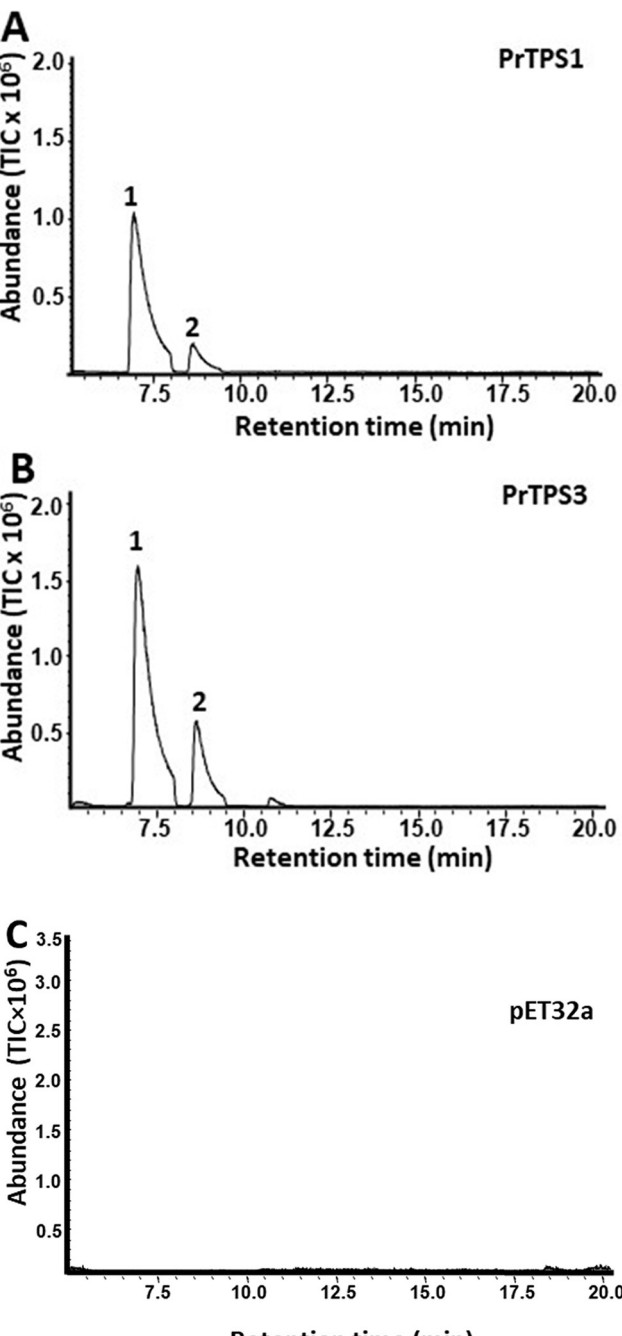

**Fig 4.** Monoterpene activity of PrTPS1 (A) and PrTPS3 (B). Crude protein extracts from heterologous expression in *E. coli* catalyzed the conversion of the substrate GPP into monoterpenes. *PrTPS* genes are identified. Products were identified by GC-MS: 1. α-pinene; 2. β-pinene. Peak numbers are consistent with those reported in Fig 1.

predominant products and δ-elemene, (+) -cycloisosativene, and δ-cadinene as the minor ones. In contrast, PrTPS7 (Fig 5C) produced germacrene D as a major product and β-ylangene as a minor product. Besides accepting (*E,E*)-FPP, PrTPS5 also converted GPP to multiple monoterpenes: α-pinene, α-terpinolene, and myrcene as predominantly detected volatile monoterpenes, and limonene, α-phellandrene, (*Z*)-β-ocimene, and (*E*)-β-ocimene as the minor ones (S2 Fig).

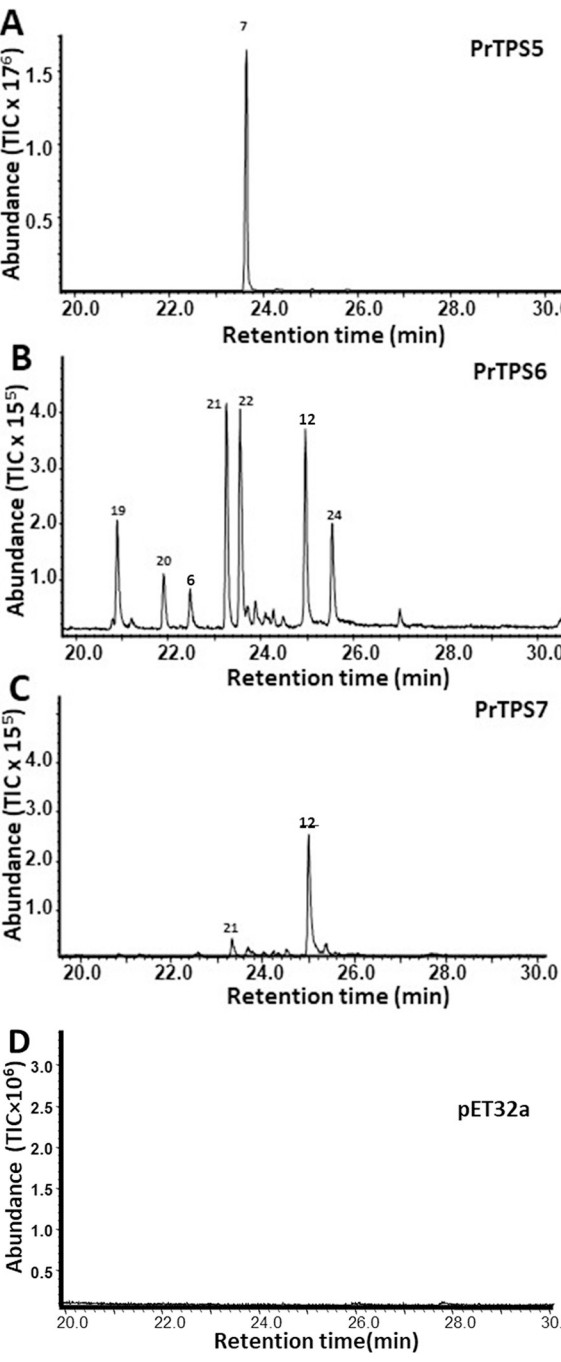

**Fig 5.** Sesquiterpene synthase activity of PrTPS5 (A), PrTPS6 (B), and PrTPS7 (C). Crude proteins extracted from heterologous expression in *E. coli* catalyzed the conversion of the substrate FPP into sesquiterpenes. Products identified by GC-MS: 6. β-elemene; 7. (E)-α-bergamotene; 12. germacrene D; 19.δ-elemene; 20. (+)-cycloisosativene; 21. β- ylangene; 22. γ-elemene; 23. δ-cadinene. Peak numbers are consistent with those reported in Fig 1.

## 3-D modelling

To gain insights into functional variations of *PrTPS*s, analyses of the 3-D models of two very resemblant *PrTPS*s, *PrTPS6* and *PrTPS7*, were accomplished in the Swiss Model. The AA identity between these two translated TPSs was 89.1% in the 548 residues overlap and gap

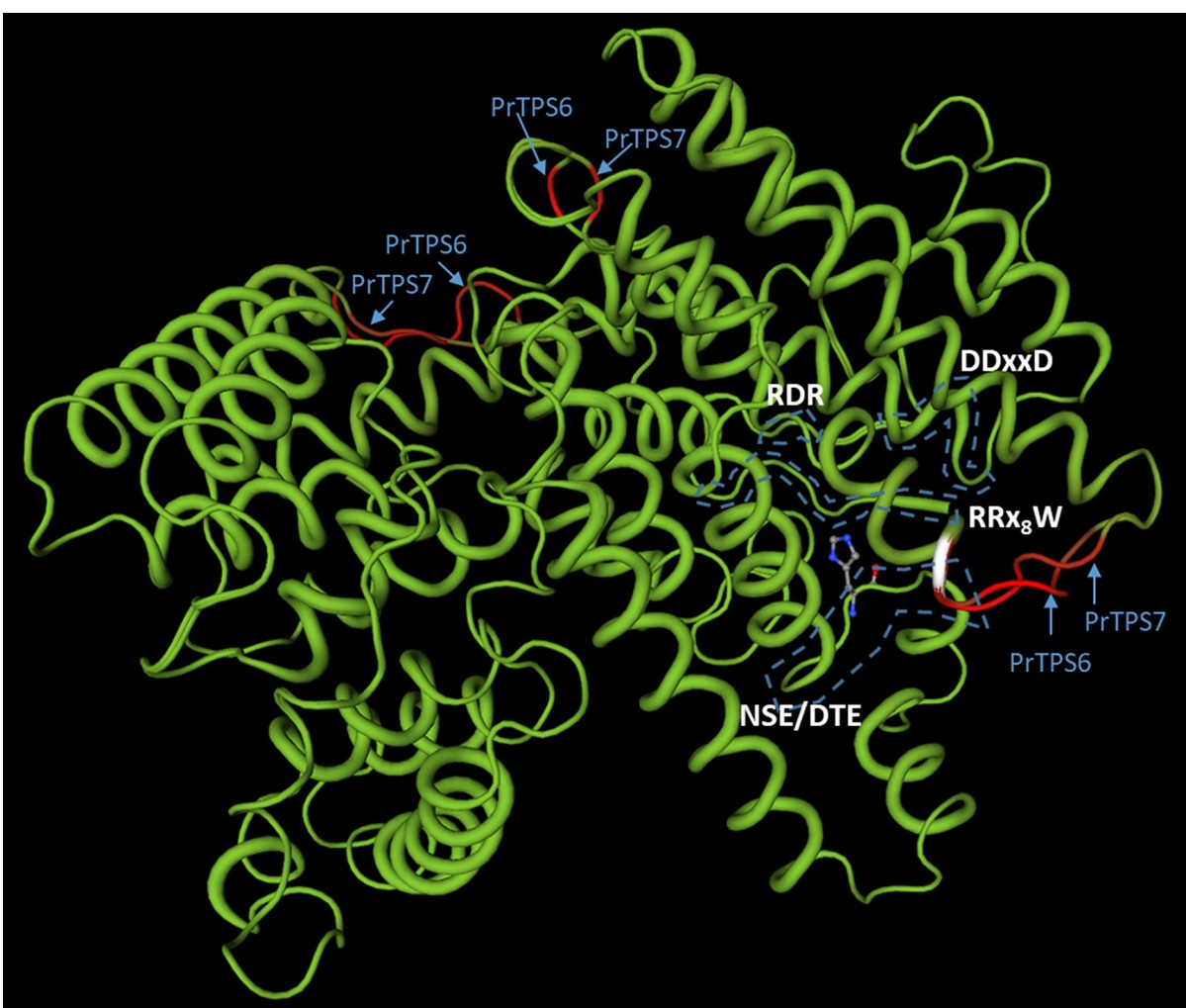

**Fig 6. Three-dimensional models of PrTPS6 and PrTPS7.** The models were computed by Swiss Model [34] using default parameters. Identical amino acid residues are shown in green; polymorphic amino acid chains are shown in red. The conserved motifs were identified using the Swiss Model server over web browser and marked per each respective coding frame.

frequency of 0.2%. The next-best similarity score among the cloned *PrTPS*s was between *PrTPS1* and *PrTPS3*, with 54.3% identity in 549 residues overlap and the gap frequency of 2.4%. From the 3 regions identified as polymorphic between *PrTPS6* and *PrTPS7* (Fig 6), the most likely region affecting the substrate specificity and activities of either *PrTPS*s is the stretch of AA placed very close to the conserved regions RRx$_8$W, NSE/DTE, RDR, and DDxxD (Fig 6; right edge). Data from the deduced AA sequences, phylogenetic analyses, and the 3-D models imply the changes in the protein's active site architecture as underlying the observed differences in the biochemical assays (see above).

## Discussion

In this study, we analyzed the terpene profiles in tissues and emitted volatiles of *P. ruthii*, an endangered Asteraceae plant. We further supported that data with the transcriptomic analyses of terpene biosynthetic pathways, followed by cloning and functional analyses of seven *PrTPS* genes with enzyme activities in terpene biosynthesis confirmed for five of them. This

information was leveraged towards the assessment of how the terpene profiles may help the plant to survive in its peculiar, highly variable ecological niche.

Terpenes, as predominant volatiles in flower scents, play important roles in plant reproductive biology by attracting pollinators or deterring florivores [37–41]. Floral terpenes, including linalool and geraniol, emit distinct aromas that attract pollinators such as bees and butterflies, thus enhancing pollination efficiency [42]. They can contribute to insect pollinator specificity by emitting specific blends of volatile compounds that attract and interact with specific pollinator species [43]. In combination with flower color, terpenes can coordinate with scent compounds to attract specific pollinators, promoting effective pollination and reproductive success [44]. Contrastingly, terpenes such as β-caryophyllene and α-pinene have been shown to deter florivores, such as herbivorous insects, by acting as repellents or causing toxicity [45].

Another important example of physiologically relevant terpenes is the abscisic acid, a terpenoid plant hormone, that plays a critical role in regulating plant responses to drought stress. It acts as a signaling molecule and helps the plants close their stomata, thereby reducing the transpiration-based water loss. Terpenes can also enhance plant water-use efficiency and improve drought tolerance by modulating plant physiology and metabolism [46]. Beyond helping the plants withstand the drought stress, terpenes have been shown to enhance plant tolerance to cold and freezing temperatures. They act as cryoprotectants by reducing the freezing point of cellular fluids, preventing ice crystal formation, and maintaining membrane stability. Additionally, terpenes can regulate the expression of genes involved in cold acclimation and promote the synthesis of protective proteins and enzymes. Monoterpenes, such as α-pinene and limonene, have been found to enhance the cold tolerance of plants by regulating the expression of cold-responsive genes and protecting the photosynthetic machinery from cold-induced damage [47]. Sesquiterpenes β-caryophyllene and α-humulene have been found to enhance freezing tolerance by reducing ice nucleation and promoting ice formation at higher subzero temperatures, protecting plant tissues from freeze-induced damage [48]. Terpenes, including isoprene and monoterpenes, have been implicated in the process of cold acclimation, enhancing the overall cold tolerance of plants by modulation of various physiological and biochemical responses [49]. Beyond freezing tolerance, terpenes are imparting salt and osmotic stress tolerance to plants: They help regulate ion transport and osmotic balance, reducing the toxic effects of high salt concentrations on plant cells [50].

Our results are consistent with the general observation that terpenes are major constituents of floral scents in many plant species [7, 42, 51–53]; this may also be the function of the floral terpenes produced by *P. ruthii* [3, 7]. α-Pinene is the predominant monoterpene detected from *P. ruthii* flowers and probably acts as an important chemical cue to attract pollinators, and similar to observations in *Eucalyptus polybractea* [54], or the moths *Helicoverpa armigera* reacting to this terpene [55]. Contrastingly, α-pinene was identified as a repellent of bee pollinators in melon flowers [56]. β-Pinene has shown potential in enhancing plant resistance against various stresses, including pathogen attack and oxidative stress [57]. Besides the pinenes, limonene was another monoterpene detected from *P. ruthii* flowers; it was previously identified as an attractant for bee pollinators [56]. Both limonene and myrcene attracted bumblebee pollinators [58].

Most terpenes, including some identified from *P. ruthii* in this study, also serve as plant chemical defense molecules [59]. They may be toxic to microbial pathogens and/or insect pests [60–62]. The variation in terpene species and concentrations in organs of *P. ruthii* plants may provide specific defenses to the various types of natural enemies they encounter, such as herbivorous insects and pathogenic fungi. For instance, during a previous reintroduction effort, we found mealybugs in many of the collected seeds samples (pers. obs.; [3]); terpenes emitted from the desiccating flower heads may have been of suboptimal composition or

concentration to repel the pest that foraged on the seeds and further endangered the plant occurrence. When plants are attacked by pests, blends of terpenes are emitted from various tissues[40, 41, 63, 64]. α-Bergamotene has been involved in plant defenses against herbivorous insects and possesses antimicrobial properties [65]. β-Elemene could be important in the establishment of mycorrhizae [66], which may aid in drought tolerance and more efficient absorption of various mineral compounds (e.g., phosphorus) from nutritionally poor sites. Furthermore, phytoalexins derived from the terpenoids in *Zea mays* roots may be associated with drought tolerance [67] or defense against biotic soilborne pathogens, including nematodes [68, 69].

Sequenced genomes in Asteraceae contain large families of terpene synthases. Many full length *TPS* genes were predicted in *Helianthus annuus* (n = 99 [70]), *Chrysanthemum nankingense* (n = 59 [53]), *C. seticuspe* (n = 66 [53]), and *Artemisia annua* (n = 88 [71]), respectively. Consistent with the *TPS* gene abundance in other Asteraceae, 67 TPS unigenes were identified in our study of *P. ruthii*. The expansion of the *TPS* gene family in many plants is the generally accepted underlying mechanism behind the diversity of the terpenoids produced on the one hand, and the neofunctionalization and spatio-temporal variability of expression of TPSs on the other hand [9, 38, 40, 72]. For the *PrTPS*s, we observed a possible duplication of *PrTPS6* and *PrTPS7* due to their unusually high sequence identity, close phylogenetic placement, and the striking overlap of their 3-dimensional models. Thus, the AA polymorphisms in the 3-D model can be underlying the detected differences in the proteins' activity, as the sequence mutation clearly affected the enzyme active site architecture. Due to lack of the high-quality genomic resources for *P. ruthii*, the claim of expansion and neofunctionalization awaits verification in future research.

Of the seven cloned putative *PrTPS*s, activity detection failed for two cloned candidates. One plausible reason could be the missing parts of ORFs, inherent to the transcriptome based ORF finding and cloning. Other reasons may be related to their expression in insoluble form or improper folding, or to the improper reaction composition including the buffering agent or pH, species and/or concentration of the metal divalent ions, or species of the substrate used [73–77]. In our study, the most abundant volatile terpenoids detected are monoterpenes α-pinene, β-pinene produced by PrTPS1 and PrTPS3, myrcene and limonene, which are the common in the floral volatile blend of genus *Chrysanthemum* [53]. Only one sesquiterpene, (*E*)-α-bergamotene produced by PrTPS5 was detected *in vivo*, and no products of the heterologously expressed *PrTPS6* and *PrTPS7* were detected neither in the extracts nor in the emitted volatiles. Their enzymatic products probably get quickly converted to other nonvolatile products; for example, volatile sesquiterpenes β-bisabolene and β-macrocarpene get readily converted to nonvolatile zealexins (β-bisabolene derivatives, β-macrocarpene derivative) by cytochrome P450 [78]. Similar was observed for many other plants including Asteraceae species: the volatile terpenes did not accumulate or get emitted, but instead were converted to more polar terpenoids. One interesting observation was that of the promiscuity of PrTPS5 that accepted both tested substrates. Such a mechanism of terpene diversity was recorded for several plants as one of the ways towards the range of products formed [74, 76]. Overall, several of the *PrTPS*s identified here may be of interest for heterologous expression and accretion of rarely occurring terpene species or whose synthesis has been thus far otherwise hampered [37, 73, 77, 79].

Isopentenyl pyrophosphate (IPP) and its isomer dimethyallyl pyrophosphate (DMAPP) are produced by MVA and MEP/DOXP pathways in plants. Compared with *C. nankingense* and *C. seticuspe*, the copy number of some of the enzymes involved in both pathways of *P. ruthii* are dramatically different. For example, AACT and HMGS had eight and six assumed isoforms, respectively, whereas only two assumed isoforms were detected each in both *C.*

*nankingense* and *C. seticuspe* [53]. This suggested that *P. ruthii* requires more copies of AACT and HMGS to ensure its supply of the basic five-carbon units for terpene biosynthesis. Furthermore, HMGR is a known rate-limiting step for the MVA pathway and the last of the initial reactions with multiple isoforms [53]. Similarities in the higher number of assumed isoforms and in the biochemical background of the reactions that involve NAPDH suggests the same rate-limiting role of DXR in the MEP/DOXP pathway for *P. ruthii*. Multiple assumed isoforms detected in the flower transcriptomes suggested a comparatively higher turnover rates in the terpene biosynthetic machinery than in the other analyzed organs that were documented in our volatile profiling. This feature also points to the metabolic flexibility and tissue/organ specific expression patterns [63], to be analyzed in follow-up studies. Finally, richness in the terpene species we detected may be related to this group of compounds playing roles in thermal and oxidative stresses. This feature may render *P. ruthii* uniquely suited to the particularly harsh environments it is most often found in ([3]; A.J. Dattilo, *unpublished*), with barely any other plants competing for the habitat.

In summary, we have documented and characterized volatile terpene chemistry in *P. ruthii*. The insights provided in our study will lay the foundation for determining the mechanisms underlying some aspects of adaptation of *P. ruthii* to a very harsh habitat that is subject to abiotic and biotic stresses. Considering the known biological functions of terpenes, the diverse and tissue- or development-specific production of monoterpenes and sesquiterpenes in *P. ruthii* suggests that they contribute to its adaptation to its niche environment. This emerging hypothesis is undergoing assessment for this species as well as for other related endangered Asteraceae plant systems.

## Supporting information

**S1 Table. *Pityopsis ruthii* transcriptomic contigs with homology to any of the 54 KEGG enzymes involved in terpene biosynthesis.** Reads for each RNA sample (s.13; s.15; s.16) and sums thereof are presented before (SEQ) and after RapClust (CLUS), respectively.
(TXT)

**S2 Table. *Pityopsis ruthii* transcriptomic normalized read counts for putative TPSs.** Reads for each RNA sample (s.13; s.15; s.16) and sums thereof are presented before (SEQ) and after RapClust (CLUS), respectively. Those are then normalized using a given contig length [bp] and expressed for before ($RPKM_i$) and after RapClust (RapClust $RPKM_i$), respectively.
(TXT)

**S3 Table. Raw data underlying the Tables 2 and 3.**
(CSV)

**S1 Fig. Alignment of translated protein sequences of 7 terpene synthases from Pityopsis ruthii cloned in this study.** Identical amino acids are highlighted with increasingly darker backgrounds. Motifs typically found in TPSs are boxed with red boarders, and the respective motif names are identified below.
(TIF)

**S2 Fig. Monoterpene synthase activity of PrTPS5.** Crude proteins extracted from heterologous expression in E. coli catalyzed the conversion of the substrate GPP into monoterpenes. Products identified by GC-MS: 1. α-pinene; 2. Myrcene; 3. α-phellandrene; 4. Limonene; 5. (Z)-β-ocimene; 6. (E)-β-ocimene; 7. α-terpinolene; 8. unidentified monoterpene.
(TIF)

## Acknowledgments

Plants were collected under Tennessee Valley Authority Permit # TE117405-2 and U.S. Fish and Wildlife Service Permit # TE134817-1.

## Author Contributions

**Conceptualization:** Feng Chen, Robert N. Trigiano.

**Data curation:** Marcin Nowicki.

**Formal analysis:** Xinlu Chen, Marcin Nowicki, Chi Zhang, Tobias G. Köllner, Miriam Payá-Milans, Matthew L. Huff.

**Funding acquisition:** Robert N. Trigiano.

**Investigation:** Xinlu Chen, Marcin Nowicki, Chi Zhang, Tobias G. Köllner, Matthew L. Huff.

**Methodology:** Xinlu Chen, Marcin Nowicki, Chi Zhang, Tobias G. Köllner, Miriam Payá-Milans, Matthew L. Huff.

**Project administration:** Feng Chen, Robert N. Trigiano.

**Resources:** Phillip A. Wadl, Feng Chen, Robert N. Trigiano.

**Software:** Marcin Nowicki, Miriam Payá-Milans, Matthew L. Huff, Margaret E. Staton.

**Supervision:** Margaret E. Staton, Feng Chen, Robert N. Trigiano.

**Validation:** Xinlu Chen.

**Visualization:** Xinlu Chen, Marcin Nowicki.

**Writing – original draft:** Xinlu Chen, Marcin Nowicki.

**Writing – review & editing:** Xinlu Chen, Marcin Nowicki, Phillip A. Wadl, Tobias G. Köllner, Miriam Payá-Milans, Matthew L. Huff, Margaret E. Staton, Robert N. Trigiano.

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
