## [Decision Letter · Decision Letter 0]

4 Apr 2023

PONE-D-23-06772Chemical profile and biosynthesis of volatile terpenes in Pityopsis ruthii, a rare and endangered flowering plantPLOS ONE

Dear Dr. Nowicki,

Thank you for submitting your manuscript to PLOS ONE. After careful consideration, we feel that it has merit but does not fully meet PLOS ONE’s publication criteria as it currently stands. Therefore, we invite you to submit a revised version of the manuscript that addresses the points raised during the review process.

We look forward to receiving your revised manuscript.

Kind regards,

Waqas Khan Kayani, PhD

Academic Editor

PLOS ONE

Journal Requirements:

   "All authors certify that they have no affiliations with or involvement in any organization or entity with any financial interest or non-financial interest in the subject matter or materials discussed in this manuscript." 

Reviewers' comments:

Reviewer's Responses to Questions

**Comments to the Author**

1. Is the manuscript technically sound, and do the data support the conclusions?

Reviewer #1: Yes

Reviewer #2: Yes

2. Has the statistical analysis been performed appropriately and rigorously? 

Reviewer #1: I Don't Know

Reviewer #2: No

3. Have the authors made all data underlying the findings in their manuscript fully available?

Reviewer #1: No

Reviewer #2: Yes

4. Is the manuscript presented in an intelligible fashion and written in standard English?

Reviewer #1: Yes

Reviewer #2: Yes

5. Review Comments to the Author

Reviewer #1: The paper entitled "Chemical profile and biosynthesis of volatile terpenes in Pityopsis ruthii, a rare and endangered flowering plant" analyzed and characterized the terpene profile of Pityopsis ruthii. In this work, the authors provided a deep analysis of the different terpenes that can be found in P. ruthii. The analytical techniques are well complemented by transcriptomic analysis, and the authors provide a list of putative genes that were tested and confirmed to be involved in terpene biosynthesis. Nevertheless, there are some major points that need to be addressed in the current version of the manuscript.

- In line 275, copy number and isoform concepts are used. But they are not the same. This should be clarified. Copy numbers = paralogs? Or isoforms of the same gene? Also, how the authors arrived to that numbers?

- The statistical analysis needs to be performed, when required. For example, in table 1, the statistical analysis will be useful to define if the levels of the specific terpenes differ significantly between tissues. A corresponding section in materials and methods needs to be added.

- In the terpene synthase activity test, the negative control is specified in materials and methods, but there is no data/reference in results. It should be specified in results (since this data is already present in the figure).

- The characterization of the new genes in P. ruthii provided novel genes involved in terpene synthesis. I strongly suggest to add a summary figure with the novel data generated (related to the pathway) at the beginning of the discussion, in order to illustrate the main findings.

- The definition of the terpene profile is, indeed, necessary to understand the physiology of P. ruthii. Nevertheless, the information regarding terpenes and biotic/abiotic stress is very superficial in the discussion. This information should be discussed in detail, with examples of other works were the role of the main terpenes found in P. ruthii have been related to stress conditions.

- The RNAseq data should be uploaded to GEO database (raw data), and the link should be provided in materials and methods.

Reviewer #2: Comments to the Author

Chemical profile and biosynthesis of volatile terpenes in Pityopsis ruthii, a rare and endangered flowering plant by Chen et.al., reported important information about a rare and endangered plant to improve its conservation efforts. This report depicts some new information regarding the biosynthesis of floral terpenes in Pityopsis ruthii which can be beneficial for chemical defense mechanisms and the attraction of pollinators. Based on the results, I speculate that the authors analyzed the terpene profiles from floral parts of the plant and studied genes involved in terpene biosynthesis. They further confirmed the involvement of these genes by studying the metabolic pathway of terpene biosynthesis. This information can be helpful in the assessment of how the terpenes may help the plant survive in different habitats.

Although this study is very important for the conservation of this plant, but only studying terpene biosynthesis is not enough to decide about the survival of the plant in different habitats. As the authors mentioned in the introduction part, this plant reached to end line because it is very tough for this plant to survive in the presence of other species. There is also no explanation that why only this specific part of the United States is promoting the growth of this plant and if we try to grow this plant in any other area, these terpene biosynthesis genes would be enough to do this. What is the purpose of studying only the terpene biosynthesis gene of already endangered plants if the plant’s growth and survivability are questioned at the end???

Title: The title is quite misleading according to the findings of the paper, as there is no biosynthesis of terpenes, instead analysis of biosynthetic pathways and genes

Abstract: The objective of the paper is not mentioned in the abstract which is an important part of the abstract. The authors only mentioned the results and conclusion.

Introduction: The introduction is good but needs to incorporate a few lines regarding the adaptation of this plant in other habitats and what else can be beneficial for their adaptation. In the introduction, the only focus is on terpene biosynthesis but there is a need to incorporate some other basic adaptation requirements. Please try to use the third form in writing papers as it is mentioned in the last paragraph of the introduction.

Materials and Methods: This section is written in detail and all the information has been provided to repeat the experiment. However no statistical analysis performed in the paper. Statistical data should provide for table 1.

Results: Well written. In every section, the authors mentioned references to support their data. As this journal has a separate discussion part, it will be good to justify your findings in the discussion section and only mention your results in the result section to avoid confusion. Quality of Figures 1 and 2 needs improvement as it is very hard to read small letters. Explain in one or two lines about the function and origin of all studied genes to make it easy to understand for the reader.

Discussion: Well written and results are well justified. Please try to use the third form in writing papers as there are multiple sentences in the discussion where authors used the first form.

Conclusion: Conclusion is missing in this paper. Authors should add a conclusion section to highlight the findings of the paper in 1 or 2 paragraphs.

6. PLOS authors have the option to publish the peer review history of their article (what does this mean?). If published, this will include your full peer review and any attached files.

Reviewer #1: No

Reviewer #2: No

---

## [Author Response · Author response to Decision Letter 0]

19 May 2023

>>Revised throughout.

 "All authors certify that they have no affiliations with or involvement in any organization or entity with any financial interest or non-financial interest in the subject matter or materials discussed in this manuscript." 

>>Corrected as per suggestion (see Cover Letter).

>> Corrected as per suggestion (see Cover Letter). Supplementary Table T3 that includes raw data was added. Statement amended to that effect.

4. We note that you have included the phrase “data not shown” in your manuscript. Unfortunately, this does not meet our data sharing requirements. PLOS does not permit references to inaccessible data. We require that authors provide all relevant data within the paper, Supporting Information files, or in an acceptable, public repository. Please add a citation to support this phrase or upload the data that corresponds with these findings to a stable repository -*+(such as Figshare or Dryad) and provide and URLs, DOIs, or accession numbers that may be used to access these data. Or, if the data are not a core part of the research being presented in your study, we ask that you remove the phrase that refers to these data.

Corrected as per suggestion (see Cover Letter). Supplementary Figure F2 added.

>>Revised throughout.

Comments to the Author 

Reviewer #1: The paper entitled "Chemical profile and biosynthesis of volatile terpenes in Pityopsis ruthii, a rare and endangered flowering plant" analyzed and characterized the terpene profile of Pityopsis ruthii. In this work, the authors provided a deep analysis of the different terpenes that can be found in P. ruthii. The analytical techniques are well complemented by transcriptomic analysis, and the authors provide a list of putative genes that were tested and confirmed to be involved in terpene biosynthesis. Nevertheless, there are some major points that need to be addressed in the current version of the manuscript.

>>Thank you for this summary.

- In line 275, copy number and isoform concepts are used. But they are not the same. This should be clarified. Copy numbers = paralogs? Or isoforms of the same gene? Also, how the authors arrived to that numbers?

>>We recognize the distinction in terminology between the in silico devised copy numbers of sequences appearing in repeated fashion, and the in vivo occurring enzymic isoforms. To help better distinguish these two, we added ‘assumed’ before ‘isoforms’, as no biochemical analyses were performed to ensure the existence of isoforms.

- The statistical analysis needs to be performed, when required. For example, in table 1, the statistical analysis will be useful to define if the levels of the specific terpenes differ significantly between tissues. A corresponding section in materials and methods needs to be added.

>>We added the results of one- and two-way ANOVA on results from Table 1 and Table 2.

- In the terpene synthase activity test, the negative control is specified in materials and methods, but there is no data/reference in results. It should be specified in results (since this data is already present in the figure).

>>Corrected in R1 – Figs 4 and 5.

- The characterization of the new genes in P. ruthii provided novel genes involved in terpene synthesis. I strongly suggest to add a summary figure with the novel data generated (related to the pathway) at the beginning of the discussion, in order to illustrate the main findings.

>>We respectfully observe, that in this report we already present 6 Figures and 2 Tables in the main text, and 2 Figures and 3 Tables in the Supplementary data. We believe that the suggested additional summary figure would dilute the message owing to the number of items presented.

- The definition of the terpene profile is, indeed, necessary to understand the physiology of P. ruthii. Nevertheless, the information regarding terpenes and biotic/abiotic stress is very superficial in the discussion. This information should be discussed in detail, with examples of other works were the role of the main terpenes found in P. ruthii have been related to stress conditions.

>>We added a thorough discussion of terpenes involvement in plant reaction to abiotic (drought; heat; cold) and biotic stresses (herbivory; pathogens), as well as to the plant reproductive biology. See ll.361-389: “Terpenes, as predominant volatiles in flower scents, play important roles in plant reproductive biology by attracting pollinators or deterring florivores[36-40]. Floral terpenes, including linalool and geraniol, emit distinct aromas that attract pollinators such as bees and butterflies, thus enhancing pollination efficiency[41]. They can contribute to insect pollinator specificity by emitting specific blends of volatile compounds that attract and interact with specific pollinator species[42]. In combination with flower color, terpenes can coordinate with scent compounds to attract specific pollinators, promoting effective pollination and reproductive success[43]. Contrastingly, terpenes such as β-caryophyllene and α-pinene have been shown to deter florivores, such as herbivorous insects, by acting as repellents or causing toxicity[44]. 

Another important example of physiologically relevant terpenes is the abscisic acid, a terpenoid plant hormone, that plays a critical role in regulating plant responses to drought stress. It acts as a signaling molecule and helps the plants close their stomata, thereby reducing the transpiration-based water loss. Terpenes can also enhance plant water-use efficiency and improve drought tolerance by modulating plant physiology and metabolism[45]. Beyond helping the plants withstand the drought stress, terpenes have been shown to enhance plant tolerance to cold and freezing temperatures. They act as cryoprotectants by reducing the freezing point of cellular fluids, preventing ice crystal formation, and maintaining membrane stability. Additionally, terpenes can regulate the expression of genes involved in cold acclimation and promote the synthesis of protective proteins and enzymes. Monoterpenes, such as α-pinene and limonene, have been found to enhance the cold tolerance of plants by regulating the expression of cold-responsive genes and protecting the photosynthetic machinery from cold-induced damage[46]. Sesquiterpenes β-caryophyllene and α-humulene have been found to enhance freezing tolerance by reducing ice nucleation and promoting ice formation at higher subzero temperatures, protecting plant tissues from freeze-induced damage[47]. Terpenes, including isoprene and monoterpenes, have been implicated in the process of cold acclimation, enhancing the overall cold tolerance of plants by modulation of various physiological and biochemical responses[48]. Beyond freezing tolerance, terpenes are imparting salt and osmotic stress tolerance to plants: They help regulate ion transport and osmotic balance, reducing the toxic effects of high salt concentrations on plant cells[49].”

- The RNAseq data should be uploaded to GEO database (raw data), and the link should be provided in materials and methods.

>>We noted already in the original submission the identity of the RNAseq data (l.265; NCBI PRJNA778727) and of the sequences of the cloned PrTPSs (l.309; GenBank accession numbers: ON166544 to ON166550).

Reviewer #2: Chemical profile and biosynthesis of volatile terpenes in Pityopsis ruthii, a rare and endangered flowering plant by Chen et.al., reported important information about a rare and endangered plant to improve its conservation efforts. This report depicts some new information regarding the biosynthesis of floral terpenes in Pityopsis ruthii which can be beneficial for chemical defense mechanisms and the attraction of pollinators. Based on the results, I speculate that the authors analyzed the terpene profiles from floral parts of the plant and studied genes involved in terpene biosynthesis. They further confirmed the involvement of these genes by studying the metabolic pathway of terpene biosynthesis. This information can be helpful in the assessment of how the terpenes may help the plant survive in different habitats.

>>Thank you for this summary.

Although this study is very important for the conservation of this plant, but only studying terpene biosynthesis is not enough to decide about the survival of the plant in different habitats. As the authors mentioned in the introduction part, this plant reached to end line because it is very tough for this plant to survive in the presence of other species. There is also no explanation that why only this specific part of the United States is promoting the growth of this plant and if we try to grow this plant in any other area, these terpene biosynthesis genes would be enough to do this. What is the purpose of studying only the terpene biosynthesis gene of already endangered plants if the plant’s growth and survivability are questioned at the end???

>>We actually thoroughly discussed the existing data on P. ruthii in this aspect already in the original submission. The following passages speak to what we currently know: ll.55-82. We note in passing, that the plant does survive when planted out of its natural environment, as our collection was maintained in pots under natural light at University of Tennessee.

Title: The title is quite misleading according to the findings of the paper, as there is no biosynthesis of terpenes, instead analysis of biosynthetic pathways and genes

>>Title updated as per Reviewer’s suggestion.

Abstract: The objective of the paper is not mentioned in the abstract which is an important part of the abstract. The authors only mentioned the results and conclusion.

>>Updated as per Reviewer’s suggestion. Ll.30-34: “Our long-term goal is to understand the mechanisms behind P. ruthii's adaptation to restricted areas in Tennessee. Here, we profiled the secondary metabolites, specifically in flowers, with a focus on terpenes, aiming to uncover the genomic and molecular basis of terpene biosynthesis in P. ruthii flowers using transcriptomic and biochemical approaches.”

Introduction: The introduction is good but needs to incorporate a few lines regarding the adaptation of this plant in other habitats and what else can be beneficial for their adaptation. In the introduction, the only focus is on terpene biosynthesis but there is a need to incorporate some other basic adaptation requirements. Please try to use the third form in writing papers as it is mentioned in the last paragraph of the introduction.

>>Please see our response to the first point raised. Ll.55-82 does contain what is known about this plant’s biology in this regard.

Materials and Methods: This section is written in detail and all the information has been provided to repeat the experiment. However no statistical analysis performed in the paper. Statistical data should provide for table 1.

>>Amended. Please also see our response to the same point raised by Reviewer 1.

Results: Well written. In every section, the authors mentioned references to support their data. As this journal has a separate discussion part, it will be good to justify your findings in the discussion section and only mention your results in the result section to avoid confusion. Quality of Figures 1 and 2 needs improvement as it is very hard to read small letters. Explain in one or two lines about the function and origin of all studied genes to make it easy to understand for the reader.

>>We only reference the data in our Results part, and References are only brought up in the Discussion part. No specific data that point to specific presented items are linked in the Discussion. Figs 1 and 2 are produced at resolution of 1100 ppi; the P1 submission system likely squished that for reviewing purposes. As for the last request, we could not gather the intent behind, but added a brief information in the Fig.2 legend.

Discussion: Well written and results are well justified. Please try to use the third form in writing papers as there are multiple sentences in the discussion where authors used the first form.

>>Thank you. This is our preference to use the first person, to signify our accountability for the data presented.

Conclusion: Conclusion is missing in this paper. Authors should add a conclusion section to highlight the findings of the paper in 1 or 2 paragraphs.

>>Conclusions part is optional in PLOS One. But, we cap the manuscript with a summary. Ll.469-476: “In summary, we have documented and characterized volatile terpene chemistry in P. ruthii. The insights provided in our study will lay the foundation for determining the mechanisms underlying some aspects of adaptation of P. ruthii to a very harsh habitat that is subject to abiotic and biotic stresses. Considering the known biological functions of terpenes, the diverse and tissue- or development-specific production of monoterpenes and sesquiterpenes in P. ruthii suggests that they contribute to its adaptation to its niche environment. This emerging hypothesis is undergoing assessment for this species as well as for other related endangered Asteraceae plant systems.”

---

## [Editor Report · Decision Letter 1]

7 Jun 2023

Chemical Profile and Analysis of Biosynthetic Pathways and Genes of Volatile Terpenes in Pityopsis ruthii, a Rare and Endangered Flowering Plant

PONE-D-23-06772R1

Dear Dr. %Nowicki%,

We’re pleased to inform you that your manuscript has been judged scientifically suitable for publication and will be formally accepted for publication once it meets all outstanding technical requirements.

Kind regards,

Waqas Khan Kayani, PhD

Academic Editor

PLOS ONE

Additional Editor Comments (optional):

Dear Marcin Nowicki,

Congratulations on the acceptance of your manuscript! I appreciate your efforts in revising the manuscript in line with the suggested revisions. The changes you made have significantly enhanced the clarity and coherence of your research. The revised manuscript now presents a more comprehensive research and I have no doubt that your work will make a valuable contribution to the academic community.

Best Regards,

Waqas Khan Kayani

PhD
---

## [Editor Report · Acceptance letter]

14 Jun 2023

PONE-D-23-06772R1 

Chemical Profile and Analysis of Biosynthetic Pathways and Genes of Volatile Terpenes in *Pityopsis ruthii*, a Rare and Endangered Flowering Plant 

Dear Dr. Nowicki:

I'm pleased to inform you that your manuscript has been deemed suitable for publication in PLOS ONE. Congratulations! Your manuscript is now with our production department. 

Kind regards, 

on behalf of

Dr. Waqas Khan Kayani 

Academic Editor

PLOS ONE